# A Brand-New Metal Complex Catalyst-Free Approach to the Synthesis of 2,8-Dimethylimidazo[1,2-*b*]pyridazine-6-Carboxylic Acid—A Key Intermediate in Risdiplam Manufacturing Process

**DOI:** 10.3390/molecules30143011

**Published:** 2025-07-18

**Authors:** Georgiy Korenev, Alexey A. Gutenev, Fyodor V. Antipin, Vladimir V. Chernyshov, Maria P. Korobkina, Maxim B. Nawrozkij, Roman A. Ivanov

**Affiliations:** Medicinal Biotechnology Department, Sirius University of Science and Technology, Olimpiyskiy Ave. 1, 354340 Sirius, Krasnodar Region, Russiavladimir.chernyshov2012@yandex.ru (V.V.C.) ivanov.ra@talantiuspeh.ru (R.A.I.)

**Keywords:** spinal muscular atrophy (SMA), risdiplam, small molecules

## Abstract

In this study, we report for the first time a brand-new protocol for the multigram-scale synthesis of 5-methyl-6-oxo-1,6-dihydropyridazine-3-carboxylic and 2,8-dimethylimidazo[1,2-*b*]pyridazine-6-carboxylic acids, without the utilization of metal-complex catalysts. The developed technology for the production of the aforementioned acids is of great importance for two reasons. Firstly, these acids serve as intermediates in the synthesis of risdiplam, the first small-molecule drug approved for the treatment of spinal muscular atrophy. Secondly, they themselves are valuable building blocks right from a broader medicinal chemistry perspective. The synthesis of risdiplam was carried out using a modified synthetic protocol, utilizing the acids indicated above as the key intermediates. The protocols presented in this study enable the production of target compounds with high purity and an acceptable yield.

## 1. Introduction

Spinal muscular atrophy, or SMA, is a rare genetic disorder that typically manifests itself in infancy or early childhood. If left undiagnosed and untreated, it is the leading genetic cause of infant mortality [1]. The condition may also arise later in life, albeit with a milder clinical course. A defining feature of spinal muscular atrophy (SMA) is significant hypotonia, presenting with symmetrical flaccid paralysis and often accompanied by poor or absent head control. Spontaneous movement is generally minimal, with antigravity limb motions rarely observed. In some cases, patients may also exhibit congenital bone fractures and exceptionally thin ribs [2,3,4,5,6]. SMA is caused by a mutation in the SMN1 gene, which encodes the SMN protein essential for the survival of motor neurons. The loss of these neurons in the spinal cord disrupts communication between the brain and skeletal muscles [7]. Another gene, SMN2, has been identified as a potential modifier of the disease, as individuals with more copies of SMN2 tend to experience a milder course of the disease [8]. The prevalence of SMA globally ranges from approximately 1:10,000 live births, with a reported carrier frequency of 1:41 in Europe and 1:51 worldwide [9,10,11]. For a long time, SMA was considered an incurable disease, however, the development of new drugs changed the lives of patients with SMA at the beginning of the 21st century [1]. Risdiplam (RG7916, RO7034067), marketed under the brand name Evrysdi^®^, was developed by F. Hoffmann-La Roche, in collaboration with PTC Therapeutics and the Spinal Muscular Atrophy Foundation. It is a pharmaceutical agent designed to manage spinal muscular atrophy (SMA), serving as a modifier of survival motor neuron 2 (SMN2)-directed RNA splicing. It was approved by the U.S. Food and Drug Administration (FDA) in August 2020 for use in adults and children aged two months or older [12].

Risdiplam was first synthesized according to the pilot scheme outlined by Ratni et al. [13], as shown in Figure 1.

This approach suffers from the poor yield of the target compound and its tedious purification. These challenges arise from the poor regioselectivity of the amination of 3,6-dichloro-4-methylpyridazine and fluorine substitution in the derivative **6** at the final step (N4 position versus N7 position, accompanied by a small amount of N4,N7-disusbtitution). Additionally, the oxidative nature of DMSO at elevated temperatures contributes to further complications during the final stage of the synthesis.

Moessner et al. published a comprehensive review [14] dedicated to the comparative analysis of various attempts for the synthesis of risdiplam during the original manufacturing campaign. The abovementioned protocol has been further optimized in an iterative manner to the ultimate commercial manufacturing process by developers (Figure 2) [14].

Extensive process understanding has been established across all stages of the synthesis, including the preparation of key starting materials, resulting in a robust and well-characterized manufacturing process supported by a comprehensive control strategy [14]. The designed commercial route affords risdiplam of high quality on a multi-kilogram scale, achieving an overall yield of 33% (Figure 2) [14].

However, while the derivatives of 2,7-disubstituted 4*H*-pyrido[1,2-*a*]pyrimidin-4-one **5**, **10** may be easily prepared by the Conrad–Limpach synthesis [15], followed by deoxychlorination or *O*-tosylation of the corresponding 2-hydroxy-compound, synthesis of 2,8-dimethyl-6-(4,4,5,5-tetramethyl-1,3,2-dioxaborolan-2-yl)imidazo[1,2-*b*]pyridazine **3** appears to be difficult. The latter compound is synthesized, in turn, via Pd(dppf)Cl_2_·CH_2_Cl_2_ complex or Pd(OAc)_2_/PCy_3_ catalyzed borylation of the 6-chloro-2,8-dimethylimidazo[1,2-*b*]pyridazine **2** (Figure 1, Figure 2, respectively).

Not many compounds are known to be an alternative to using 2,8-dimethylimidazo[1,2-*b*]pyridazine derivatives as building blocks in the risdiplam synthesis. There are several known ways to obtain risdiplam utilizing ethyl 3-(2,8-dimethylimidazo[1,2-*b*]pyridazin-6-yl)propiolate [16] **12** and 2,8-dimethylimidazo[1,2-*b*]pyridazine-6-carboxylic acid [17] **13** instead of 2,8-dimethyl-6-(4,4,5,5-tetramethyl-1,3,2-dioxaborolan-2-yl)imidazo[1,2-*b*]pyridazine **3** as the key-intermediates (Figure 3).

However, the synthesis of derivatives **12**, **13** is also carried out via a metal-complex catalyzed reactions, i.e., coupling of compound **2** with ethyl propiolate in the presence of CuI, Pd[(PPh_3_)_4_] or carbonylation of compound **2** in the presence of Pd(dppp)Cl_2_, respectively (Figure 3).

The last compound **2**, however, seems to be an obstacle in practically all known risdiplam synthetic routes. Looking simple from the very first glance, this compound is very tedious to prepare and isolate in a pure state, free from the positional isomers. In fact, as was described before, it may be prepared by the heterocyclization of bromoacetone, chloroacetone, or 2-methoxyallylbromide (generated in situ) with 6-chloro-4-methylpyridazin-3-amine. The last one could be obtained either by direct ammonolysis of 3,6 dichloro-4-methylpyridazine **1** (Figure 1) (typically, in a hard-to-separate mixture with regioisomeric 3-chloro-4-methylpyridazin-6-amine) or by a hard-to-perform cascade of bromination and modified Negishi-coupling, starting from 3-amine-6-chloropyridazine [14].

Prior to comparing all known risdiplam synthetic pathways, it is necessary to outline that all of them, more or less, depend on the palladium complex catalysis. Moreover, the hard-to-regenerate expensive catalysts, toxic organochlorine solvents (the stable environmental pollutants), strict methane formation control, and huge amounts of waste may be reasonably considered the main disadvantages both from the points of green chemistry and technology scale-up.

Thus, the present study is devoted, firstly, to developing an original and convenient protocol for the 2,8-dimethylimidazo[1,2-*b*]pyridazine-6-carboxylic acid **13** synthesis, both as a known risdiplam intermediate and an attractive building block for different medicinal chemistry purposes, without the use of metal-catalyzed reactions. Secondly, our study is aimed at modifying the known methods of risdiplam synthesis, starting from the acid **13**.

## 2. Results and Discussion

### 2.1. 2,8-Dimethylimidazo[1,2-b]pyridazine-6-carboxylic Acid Synthesis

The cornerstone of this work is the original and inexpensive procedure for the synthesis of 2,8-dimethylimidazo[1,2-*b*]pyridazine-6-carboxylic acid **13**, which is, in fact, a key intermediate in the risdiplam synthesis. This makes it different from the already published risdiplam synthesis schemes [13,14,16,17].

The synthesis of the target acid **13** started from commercially available methyl pyruvate **14**, which, in the presence of Na_2_CO_3_, was converted into 5-methoxy-4-methyl-2,5-dioxopent-3-enoic acid **15** with a 40% yield (Figure 4) [18]. Subsequent cyclocondensation of acid **15** with N_2_H_4_ hydrate resulted in the hydrazine salt of 5-methyl-6-oxo-1,6-dihydropyridazine-3-carboxylic acid **16** formation with a 62% yield, which was further converted in 5-methyl-6-oxo-1,6-dihydropyridazine-3-carboxylic acid **17** with an 86% isolated yield (a 21% overall yield over 3 steps) (Figure 4). It is worth noting that there are currently not many protocols described for the acid **17** synthesis, and the one presented in this article is perhaps the most technologically advanced [19]. The sequential transformation of acid **17** into the corresponding ethyl ester **18** (87% isolated yield) and the deoxychlorination reaction [20] of ester **18** with POCl_3_ (88% isolated yield) resulted in the formation of ethyl 6-chloro-5-methylpyridazine-3-carboxylate **19**. The nucleophilic substitution of the chlorine atom in compound **19** for the azido group resulted in compound **20** (existing primarily as tetrazolo[1,5-*b*]pyridazine tautomer) [21,22,23] with a 92% yield. The subsequent reduction with Cu powder [24] in glacial AcOH led to the ethyl 6-amino-5-methylpyridazine-3-carboxylate **21** formation with an 85% isolated yield (Figure 4).

There are several important points to note. First, while elaborating this synthetic pathway, a key challenge was to preserve the ester functional group while achieving regioselective amination at the C6 position, ensuring technological feasibility and eliminating the need for chromatographic separation of the potential byproducts. Taking into consideration the impracticality of employing primary or secondary amines for direct amination, the target compound **21** was, instead, obtained via the reduction of the corresponding ethyl-6-azido-5-methylpyridazine-3-carboxylate **20**. The synthesis of the above-mentioned azide precursor **20** was initially attempted using ethyl 6-(tosyloxy)-5-methylpyridazine-3-carboxylate, as a starting material for the reaction with NaN_3_. However, this strategy proved to be ineffective, as the reaction predominantly led to the nucleophilic attack at the S-atom, resulting in the formation of tosyl azide and recovery of the starting material **18**. A successful approach was ultimately developed through a deoxychlorination reaction using phosphorus oxychloride, followed by nucleophilic substitution with sodium azide, enabling the efficient generation of the desired intermediate **20**. However, a seemingly simple process for the reduction of compound **20** to ethyl 6-amino-5-methylpyridazine-3-carboxylate **21** also turned out to be a non-trivial task.

The classical approach to azide reduction involves the Staudinger reaction [25], followed by hydrolysis of the obtained iminophosphorane derivative. To streamline these steps, the reaction was conducted in glacial AcOH. However, UPLC-MS analysis revealed that the reaction proceeded only at the reflux conditions, leading to an inevitable formation of the undesired ethyl 6-acetamido-5-methylpyridazine-3-carboxylate by-product in a 1:1 ratio with a desired compound **21** (Figure 5). Moreover, Ph_3_PO, which is notoriously hard to remove from the reaction mixture, has been formed as a by-product [26]. Alternative reduction methods, including H_2_ (Pd/C), NaBH_4_/MeOH, and Na_2_S/EtOH failed to yield the desired product. Utilizing Zn dust in glacial AcOH led to the unexpected formation of ethyl 8-methyl-5,6-dihydrotetrazolo[1,5-*b*]pyridazine-6-carboxylate **20.1**. It happened**,** likely, due to the predominant existence of the starting material as a tetrazolo[1,5-*b*]pyridazine tautomer (Figure 5).

The few examples presented in the scientific and patent literature refer to Ulomskii and colleagues [24], who previously demonstrated that such systems undergo efficient reduction in the Cu/glacial AcOH conditions. It is noteworthy, that the temperature regime (80–85 °C) during the reduction of compound **20** is crucial to the formation of the target product **21** in a good yield, avoiding the formation of 6-acetamido-5-methylpyridazine-3-carboxylate as a by-product.

The final stages of acid **13** synthesis included cyclocondensation of ethyl 6-amino-5-methylpyridazine-3-carboxylate **21** with chloroacetone, which resulted in ethyl 2,8-dimethylimidazo[1,2-*b*]pyridazine-6-carboxylate **22** with an 85% isolated yield. Subsequent alkaline hydrolysis allowed one to obtain the desired 2,8-dimethylimidazo[1,2-*b*]pyridazine-6-carboxylic acid with a 90% isolated yield (Figure 4). Thus, the desired intermediate **13** can be obtained from compound **17** through a six-step synthetic sequence with an overall yield of 46%, which is 15% higher than the reported yields for compound **2** via the optimized procedures developed by the original authors [14]. Moreover, the approach presented in this work enables the synthesis of the starting material **17**—albeit in moderate yield—from inexpensive methyl pyruvate (approximately $30/kg) and commonly available laboratory reagents. This is particularly noteworthy given the catalog price of compound **17** exceeding $1000/g (Enamine), and the developed route allows its preparation on the scale of several hundred grams from 1 kg of methyl pyruvate.

### 2.2. Modified Scheme for Risdiplam Synthesis

An important part of our work was the synthesis of risdiplam from 2,8-dimethylimidazo[1,2-*b*]pyridazine-6-carboxylic acid **13**. The 2nd generation process (“Meldrum’s acid route to risdiplam”) developed by Roche [14], which was eventually abandoned due to the absence of stable intermediates amenable to crystallization, served as a starting point for this protocol. These concerns have been considered a key disadvantage of this process for future scale-up and the establishment of a robust impurity control strategy. The process developed by Roche [14] included the following successive stages.

Meldrum’s acid has been acylated under basic conditions with 2,8-dimethylimidazo[1,2-*b*]pyridazine-6-carbonylchloride, prepared in situ from the corresponding acid **13**. The obtained enol-compound has been subjected to a deoxychlorination reaction, yielding a corresponding vinylchloride derivative. A 1,4 addition/elimination reaction of the latter with tert-butyl 7-(6-aminopyridin-3-yl)-4,7-diazaspiro[2.5]octane-4-carboxylate **28**, followed by heterocyclization, led to the N-Boc-protected carboxy-risdiplam. It finally underwent an acid-catalyzed Boc-deprotection/decarboxylation to give the target risdiplam trihydrate in about a 55% overall yield (from the 2,8-dimethylimidazo[1,2-*b*]pyridazine-6-carboxylic acid **13**) [14].

Our approach to synthesis underwent substantial modifications compared to the previously established method (Figure 6).

At the first stage, acylchloride **23** was obtained in an almost quantitative yield according to a known protocol [27], with a significantly reduced amount of DMF and the addition of an equimolar amount of DIPEA to ensure reproducible intermediate solubility.

The subsequent steps of synthesis were carried out as a “one-pot” procedure. The conditions for obtaining adduct **24**, in fact, did not differ a lot from the originator [14]—only an excess of DMAP was reduced to the catalytic amount in the reaction with Meldrum’s acid. Notably, the preparation of adduct **25** has been documented in the literature only once by its original developer, exclusively utilizing oxalyl chloride as the deoxychlorinating agent. Our research demonstrates that the same adduct **25** can be successfully obtained using triphosgene, following the protocols outlined by Saputra and colleagues, with pyridine substituted by DMAP as the required base [28]. It is noteworthy, that triphosgene is more stable, less corrosive, and solid at RT, in contrast to oxalyl chloride. This approach enables the elimination of the poorly defined “titration” process described initially [14] while maximizing conversion efficiency.

Further interaction of the vinylchloride derivative **25** with amine **28** was carried out under conditions similar to the published data [14], with the exception of replacing Bu_3_N with DIPEA. At this point, the reaction mixture has already accumulated a substantial amount of salts, impurities, and unreacted intermediates. The subsequent steps, which do not introduce new structural fragments but require heating, pose challenges in monitoring impurity build-up and reaction completion—key concerns highlighted by developers of the initial synthetic route. To overcome these issues, solvent extraction with phase replacement is proposed as a critical refinement. In the case of DCM as the organic solvent, a stable emulsion has been formed; however, substituting it with MTBE in combination with NH_4_Cl_aq_ enables the quantitative isolation of the target adduct **26** (with a 71% yield over 4 steps). This intermediate being a stable crystalline product, serves as a crucial control point, ensuring process robustness.

The cyclization of the adduct **26** to compound **27** was carried out by heating the reaction mixture at 95 °C in *n*-butanol, instead of reflux in *n*-propanol as described before (Figure 6) [14]. The Boc-protective group removal with the following decarboxylation of compound **27** was carried out with AcCl in *n*-butanol to form a mixture of risdiplam dihydrochloride and corresponding carboxy-risdiplam dihydrochloride. The efficiency of decarboxylation is determined by the concentration of HCl in *n*-butanol, which decreases with heating. To sustain the required acidity, portions of AcCl in *n*-butanol are gradually added.

Risdiplam dihydrochloride is eventually formed with a 55.4% yield over six steps (from the starting acid **13**). The subsequent risdiplam conversion in the trihydrate form was carried out by neutralization with NaOH in an aqueous EtOH solution with an 80% yield by the stage and with an overall 44% yield starting from the acid **13** (Figure 6).

Synthesis of the *tert*-butyl 7-(6-aminopyridin-3-yl)-4,7-diazaspiro[2.5]octane-4-carboxylate **28** was carried out in a similar way to the original protocol [14] with one exception—4,7-diazaspirodiamine[2.5]octane **29** was used as the starting material instead of *tert*-butyl 4,7-diazaspiro[2.5]octane-4-carboxylate (Figure 7).

To obtain the desired protected regioisomer, the less sterically hindered N-atom of compound **29** was protected by a Trt-group, after which the Boc-protecting group was regioselectively introduced, and the Trt-group was removed with oxalic acid to form oxalate **30** (Figure 7). Subsequent nucleophilic substitution of the bromine atom in 5-bromo-2-nitropyridine, followed by reduction of the resulting compound **31** led to the desired amine **28** (Figure 7).

Thus, the proposed protocol allows one to obtain an extremely valuable compound 2-(2,8-dimethylimidazo[1,2-*b*]pyridazin-6-yl)-7-(4,7-diazaspiro[2.5]octan-7-yl)-4*H*-pyrido[1,2-*a*]pyrimidin-4-one—the active substance of the drug Evrysdi^®^—the first oral medication approved to treat SMA, using cheap and accessible starting materials without using metal-catalyzed cross-coupling reactions and additional chromatography purification.

## 3. Materials and Methods

### 3.1. General

All solvents and reagents were obtained from commercial sources and used without further purification unless otherwise stated. ^1^H and ^13^C NMR spectra were recorded on a Bruker Avance Neo 400 MHz spectrometer (Bruker, Ettlingen, Germany, 400.1, 100.6 MHz, respectively)) in CDCl_3_, DMSO-*d*_6_ solutions. Chemical shifts δ are reported in parts per million (ppm); multiplicity: *s*, singlet; *d*, doublet; *t*, triplet; *q*, quartet; *dd*, double of doublets; *m*, multiplet; *br*, broad; the coupling constants *J* are reported in Hz. The structure of the products was determined by analyzing ^1^H and ^13^C NMR spectra; and assignments on a routine basis by a combination of 1D and 2D experiments (HSQC, HMBC). UPLS-MS analyses were performed on a «Vanquish Flex» chromatograph (Thermo Scientific, Waltham, MA, USA) with Diode Array Detector FG (DAD FG, Thermo Scientific, USA) combined with an ISQ EM Single Quadrupole Mass Spectrometer (Thermo Scientific, USA). A 6-minute gradient separation on an Agilent Poroshell 120 EC-C18 (Agilent Technologies, Santa Clara, CA, USA, 100 mm·2.0 mm, particle size 1.9 μm) column was run under the following conditions: solvent A = water with 0.1% formic acid, solvent B = acetonitrile with 0.1% formic acid, from 0 to 2.5 min—gradient elution from A:B = 9:1 to A:B = 1:9, from 2.5 min to 3.5 min—elution in A:B = 1:9, from 3.5 min to 6 min—equilibration of the chromatographic column in A:B = 19:1 at a flow rate 0.5 mL/min. The column temperature was 40 °C, injection volume of the sample was 1 µL. An electrospray ionization source was used to ionize the samples. Ions of positive and negative polarities were detected in the full ion current recording mode; the range of recorded masses was 10–700 *m/z*. The absorption spectra were recorded on a diode array detector at 2 wavelengths: 220 nm and 254 nm. High-resolution mass spectrometry (HRMS) analyses were performed using a Bruker maXis II 4G ETD mass spectrometer (Bruker, Ettlingen, Germany) and an UltiMate 3000 chromatograph (Thermo Scientific, Waltham, MA, USA) equipped with Acclaim RSLC 120 C18 2.2 μm 2.1·100 mm column (Thermo Scientific, Waltham, MA, USA). Spectrum registration mode was electrospray ionization (ESI), with a full scan between *m/z* 100 and 1500, tandem MS (MS/MS) with selection of the three most intense ions, collision-induced dissociation (CID) at 10–40 eV, and nitrogen as a collision gas. Melting points were determined on Melting Point Apparatus SMP50 (Norrscope, Bicknacre, UK) in the following regime (1 °C per minute). The target substances were lyophilized using a LABCONCO FreeZone 2.5 l freeze dryer (Labconco, Kansas City, MO, USA) samples were preliminarily frozen in a freezer at −80 °C for 5 h, sublimation was carried out for 12 h, residual pressure 0.003 mbar). Thin-layer chromatography (TLC) was carried out on Merck silica gel 60 F254 precoated plates (Merck, Rahway, NJ USA); compounds on TLC were visualized with UV light (254 nm) or by ninhydrin or phosphomolybdic acid staining. Column chromatography was performed on silica gel (60–200 mesh, Sisco, Mumbai, India).

### 3.2. Synthetic Procedures and Compounds Characterization

#### 3.2.1. Synthesis of 5-Methoxy-4-methyl-2,5-dioxopent-3-enoic Acid **15** [18]

A well-stirred suspension of Na_2_CO_3_ (1570.9 g, 14.82 mol, 2 eq) in MeOH (11.2 L) was treated by the dropwise addition of methyl pyruvate (670 mL, 7.41 mol, 1 eq) at 50 °C and then refluxed for 3 h with constant stirring. During this time, the reaction mixture became green and then yellow. After the reaction was completed [TLC (*n*-C_6_H_14_:EtOAc = 1:1, by volume, *R_f_* = 0.55 for intermediate lactone, UV; eluent—DCM:MeOH = 3:1, by volume, *R_f_* = 0.22 for the reaction product, UV], the mixture was cooled down to 20–22 °C and filtered. The filter cake was washed with ice-cold MeOH and the combined filtrate was stripped down from the solvent under the diminished pressure. The resulting residue was dissolved in deionized water (400 mL), acidified with concentrated HCl to pH ≈ 1, and extracted with MTBE (3 × 400 mL). The combined organic extracts were washed with brine (350 mL), dried over anhydrous Na_2_SO_4_ for 1 h with vigorous stirring, filtered, and evaporated in vacuo to yield a technical 5 methoxy-4-methyl-2,5-dioxopent-3-enoic acid as a dark orange viscous oil (255 g, 40%), which was used further without additional purification.

#### 3.2.2. Synthesis of 5-Methyl-6-oxo-1,6-dihydropyridazine-3-carboxylic Acid **17**

A well-stirred solution of technical 5-methoxy-4-methyl-2,5-dioxopent-3-enoic acid (252 g, 1.465 mol, 1 eq) from the previous step in MeOH (2 L) was treated dropwise with 50% aqueous N_2_H_4_ (180 mL, 2.7832 mol, 1.9 eq). During the addition, an exothermic reaction took place, resulting in the formation of a heavy precipitate and a red supernatant solution. The reaction mixture was stirred for an additional 1 h, and gradually cooled in an ice bath until the complete crystallization of the resulting salt. The latter was filtered off and the filter cake was washed with ice-cold MeOH (150 mL) and dried at reduced pressure to a constant weight to yield 169 g (62%) of hydrazine salt of 5-methyl-6-oxo-1,6-dihydropyridazine-3-carboxylic acid. The obtained salt was dissolved in deionized water (450 mL) at 70 °C and the resulting solution was acidified to pH ≈ 1 with concentrated HCl. The resulting suspension was gradually cooled in an ice bath until the complete crystallization of the resulting 5-methyl-6-oxo-1,6-dihydropyridazine-3-carboxylic acid and filtered. The filter cake was washed with cold deionized water (2 × 100 mL), dried at reduced pressure, and suspended in toluene (100 mL). The mixture was stripped down from the solvent in vacuo (45 °C, 4 mbar, 2.5 h) to yield 120 g (86%) of the target 5-methyl-6-oxo-1,6-dihydropyridazine-3-carboxylic acid.

^1^H NMR (400 MHz, DMSO-*d_6_*, *δ*): 13.36 (1H, *s*, 1-NH), 7.74 (1H, *q*, *J* = 1.2 Hz, 4-CH), 2.09 (3H, *d*, *J* = 1.3 Hz, 5-CH_3_). ^13^C NMR (101 MHz, DMSO-*d_6_*, *δ*): 164.4 (3-COOH), 162.3 (C-6), 139.9 (C-5), 137.4 (C-3), 129.4 (C-4), 16.2 (5-CH_3_). HRMS (ESI–): found *m/z* 153.0304 [M − H]^−^; calculated for C_6_H_5_N_2_O_3_ 153.0378, mp = 256–257 °C (dec).

#### 3.2.3. Synthesis of Ethyl 5-Methyl-6-oxo-1,6-dihydropyridazine-3-carboxylate **18**

A stirred suspension of 5-methyl-6-oxo-1,6-dihydropyridazine carboxylic acid (100 g, 0.649 mol, 1 eq) in EtOH (800 mL), containing a catalytic amount of conc. H_2_SO_4_ (3.5 mL, 64.9 mmol, 0.1 eq) was kept at reflux for 4 h, during which time a clear solution was formed. After completion of the reaction (TLC, eluent—*n*-BuOH:H_2_O:AcOH, 4:1:1, by volume, UV-detection), the whole mixture was kept at approximately +4 °C for 12 h and filtered. A filter cake was washed with ice-cold EtOH (150 mL) and dried under reduced pressure. The filtrate was evaporated in vacuo*,* diluted with deionized water (300 mL), and neutralized with saturated NaHCO_3_ solution (150 mL). The residual target ethyl 5-methyl-6-oxo-1,6-dihydropyridazine-3-carboxylate was additionally extracted with MTBE (3 × 150 mL), the combined organic extracts were washed with brine (200 mL), dried over anhydrous Na_2_SO_4_ for 1 h with vigorous stirring, filtered, and evaporated in vacuo. The combined portions of grayish powder of ethyl 5-methyl-6-oxo-1,6-dihydropyridazine-3-carboxylate were dried under reduced pressure until a constant weight (102.8 g, 87%).

^1^H NMR (400 MHz, DMSO-*d_6_*, *δ*): 13.43 (1H, *s*, 1-NH), 7.75 (1H, *q*, *J* = 1.4 Hz, 4-CH), 4.30 (2H, *q*, *J* = 7.1 Hz, CH_2_CH_3_), 2.10 (3H, *d*, *J* = 1.3 Hz, 5-CH_3_), 1.30 (3H, *t*, *J* = 7.1 Hz, CH_2_CH_3_). ^13^C NMR (101 MHz, DMSO-*d_6_*, *δ*): 162.8 (3-COOCH_2_CH_3_), 162.2 (C-6), 140.2 (C-5), 136.7 (C-3), 129.1 (C-4), 61.9 (3-COOCH_2_CH_3_), 16.2 (3-COOCH_2_CH_3_), 14.5 (5-CH_3_). HRMS (ESI+): found *m/z* 183.0793 [M + H]^+^; calculated for C_8_H_11_N_2_O_3_ 183.0691, mp = 169–170 °C.

#### 3.2.4. Synthesis of Ethyl 6-Chloro-5-methylpyridazine-3-carboxylate **19**

A mixture of ethyl 5-methyl-6-oxo-1,6-dihydropyridazine-3-carboxylate (102.8 g, 0.565 mol, 1 eq) and POCl_3_ (360 mL, 3.85 mol, 6.8 eq) was vigorously stirred at 50–55 °C (bath temperature) for 3.5 h. Upon the completion of the reaction (TLC, eluent—*n*-C_6_H_14_:EtOAc = 1:1, by volume, *R_f_* = 0.33 for the starting compound, *R_f_* = 0.73 for the reaction product, UV-detection), the resulting solution was stripped down from the solvent under reduced pressure and the residue was dissolved in MTBE (500 mL). The resulting solution was poured into 1 L of ice-cold saturated NaHCO_3_ solution with vigorous stirring. The organic phase was separated and the aqueous layer was extracted with MTBE (2 × 300 mL). The combined organic layers were washed sequentially with deionized water (300 mL) and brine (300 mL), dried over anhydrous Na_2_SO_4_ for 1 h with vigorous stirring, and filtered. The filtrate was evaporated to dryness in vacuo to give ethyl 6-chloro-5-methylpyridazine-3-carboxylate as a viscous brown oil (113.2 g, 88%).

^1^H NMR (400 MHz, DMSO-*d_6_*, δ): 8.28 (1H, *q*, *J* = 0.9 Hz, 4-CH), 4.43 (2H, *q*, *J* = 7.1 Hz, CH_2_CH_3_), 2.47 (3H, *d*, *J* = 0.9 Hz, 5-CH_3_), 1.37 (3H, *t*, *J* = 7.1 Hz, CH_2_CH_3_). ^13^C NMR (101 MHz, DMSO-*d_6_*, δ): 163.6 (3-COOCH_2_CH_3_), 160.2 (C-6), 151.2 (C-3), 140.1 (C-5), 131.0 (C-4), 62.6 (3-COOCH_2_CH_3_), 19.0 (5-CH_3_), 14.5 (3-COOCH_2_CH_3_). HRMS (ESI+): found *m/z* 201.0452 [M + H]^+^; calculated for C_8_H_10_^35^ClN_2_O_2_ 201.0353.

#### 3.2.5. Synthesis of Ethyl 8-Methyltetrazolo[1,5-*b*]pyridazine-6-carboxylate (or Ethyl 6-Azido-5-methylpyridazine-3-carboxylate as Its Tautomeric Form **20**)

A solution of ethyl 6-chloro-5-methylpyridazine-3-carboxylate (113.2 g, 0.565 mol, 1 eq) in anhydrous DMF (225 mL) was treated with NaN_3_ (40.4 g, 0.621 mol, 1.1 eq) and stirred at 70-75 °C for 3 h. Upon the completion of the reaction (TLC, eluent—*n*-C_6_H_14_:EtOAc = 1:1, by volume, *R_f_* = 0.72 for the starting compound, *R_f_* = 0.49 for the reaction product, UV-detection), the resulting mixture was stripped down from the solvent at reduced pressure, the residue was then treated with MTBE (600 mL) and washed with deionized water (3 × 200 mL). The organic phase was separated and the aqueous washings were extracted with MTBE (2 × 200 mL). The combined organic extracts were washed with 250 mL of brine, dried over anhydrous Na_2_SO_4_ for 1 h with vigorous stirring, and filtered. The organic filtrate was evaporated to dryness in vacuo to give the crude ethyl 6-azido-5-methylpyridazine-3-carboxylate as a red oil, which, in turn, dissolved in 200 mL of a mixture of *n*-C_6_H_14_ and MTBE (3:2, by volume). Upon vigorous stirring, the resulting solution deposited a crop of fine crystalline material. An additional amount of the crystals was obtained by chilling this mixture at −20 °C for 12 h. The precipitate was filtered off, washed sequentially with 100 mL of an ice-cold mixture of *n*-C_6_H_14_ and MTBE (3:1, by volume), and then—with 200 mL of *n*-C_6_H_14_, dried at reduced pressure to the constant weight, giving the ethyl 6-azido-5-methylpyridazine-3-carboxylate (107.8 g, 92%) as a dark red crystalline powder.

^1^H NMR (400 MHz, DMSO-*d_6_*, *δ*): 8.16 (1H, *q*, *J* = 1.2 Hz, 4-CH), 4.49 (2H, *q*, *J* = 7.1 Hz, CH_2_CH_3_), 2.82 (3H, *d*, *J* = 1.2 Hz, 5-CH_3_), 1.41 (3H, *t*, *J* = 7.1 Hz, CH_2_CH_3_). ^13^C NMR (101 MHz, DMSO-*d_6_*, *δ*): 162.0 (3-COOCH_2_CH_3_), 147.2 (C-3), 145.3 (C-6), 139.7 (C-5), 124.2 (C-4), 63.4 (3-COOCH_2_CH_3_), 17.0 (5-CH_3_), 14.4 (3-COOCH_2_CH_3_). HRMS (ESI+): found *m/z* 208.0849 [M + H]^+^; calculated for C_8_H_10_N_5_O_2_ 208.0756, mp = 107–108 °C.

#### 3.2.6. Synthesis of Ethyl 6-Amino-5-methylpyridazine-3-carboxylate **21**

A vigorously stirred solution of ethyl 8-methyltetrazolo[1,5-*b*]pyridazine-6-carboxylate (tautomer of ethyl 6-azido-5-methylpyridazine-3-carboxylate) (99.5 g, 0.481 mol, 1 eq) in glacial AcOH (400 mL) was treated with the portionwise addition of the copper powder (40 g, 625 mg-atom, 1.3 eq) at 80–85 °C during a 6 h period. Caution! The procedure is accompanied by vigorous nitrogen evolution! After the completion of the reaction (TLC, eluent—*n*-C_6_H_14_:EtOAc = 1:1, by volume, *R_f_* = 0.49 for the starting material, UV-detection; eluent—EtOAc, *R_f_* = 0.13 for target product, UV-detection or ninhydrin visualization), the solvent was distilled off in vacuo, the residue was treated with EtOH (200 mL) and filtered through a pad of Celite^®^. A filter cake was washed with EtOH (2 × 50 mL). The combined filtrate was added into the vigorously stirred cold mixture of 30% aqueous NH_3_ solution (400 mL) and deionized water (2.5 L), leading to the formation of a voluminous precipitate, which was filtered off, washed with cold deionized water, and dried at reduced pressure. The target ethyl 6-amino-5-methylpyridazine-3-carboxylate was obtained as a brown crystalline powder (73.95 g, 85%).

^1^H NMR (400 MHz, DMSO-*d_6_*, *δ*): 7.66 (1H, *d*, *J* = 1.2 Hz, 4-CH), 6.99 (2H, *br.s.*, 6-NH_2_), 4.31 (2H, *q*, *J* = 7.1 Hz, CH_2_CH_3_), 2.11 (3H, *d*, *J* = 1.0 Hz, 5-CH_3_), 1.32 (3H, *t*, *J* = 7.1 Hz, CH_2_CH_3_). ^13^C NMR (101 MHz, DMSO-*d_6_*, *δ*): 164.8 (3-COOCH_2_CH_3_), 162.1 (C-6), 143.2 (C-3), 128.3 (C-4), 122.4 (C-5), 61.1 (3-COOCH_2_CH_3_), 16.7 (5-CH_3_), 14.7 (3-COOCH_2_CH_3_). HRMS (ESI+): found *m/z* 182.0956 [M + H]^+^; calculated for C_8_H_12_N_3_O_2_ 182.0851, mp = 155–156 °C.

#### 3.2.7. Synthesis of Ethyl 8-Methyl-5,6-dihydrotetrazolo[1,5-*b*]pyridazine-6-carboxylate **20.1**

A solution of ethyl 8-methyltetrazolo[1,5-*b*]pyridazine-6-carboxylate (tautomer of ethyl 6-azido-5-methylpyridazine-3-carboxylate) (0.5 g, 2.42 mmol, 1 eq) in a mixture of dry THF (8 mL) and glacial AcOH (2 mL) was cooled to 0 °C and treated with Zn dust (634 mg, 9.7 mmol, 4 eq). The reaction mixture was stirred at 0 °C for 2 h. Upon the complete conversion of the starting pyridazine derivative (TLC, eluent—n-C_6_H_14_:EtOAc = 1:1, by volume, *R_f_* = 0.6 for the starting compound, *R_f_* = 0.3 for the target product), the reaction mixture was quenched with a mixture of saturated aqueous solution of NaHCO_3_ (20 mL) and MTBE (10 mL) and filtered. The phases were separated and the aqueous layer was extracted with MTBE (2 × 15 mL). The combined organic extracts were washed with 20 mL of brine, dried over anhydrous Na_2_SO_4_, and filtered. The resulting filtrate was evaporated to dryness under diminished pressure to give a portion of crude ethyl 8-methyl-5,6-dihydrotetrazolo[1,5-*b*]pyridazine-6-carboxylate as an orange oil, which was dissolved in EtOAc (2 mL) and re-precipitated with *n*-hexane (5 mL), yielding an orange powder. The latter was filtered off, washed with Et_2_O (7 mL), and air-dried to a constant weight (130 mg, 25%).

^1^H NMR (400 MHz, DMSO-*d_6_*, *δ*): 7.93 (1H, *d*, *J* = 6.8 Hz, 5-NH), 6.36–6.33 (1H, *m*, 7-CH), 5.03-5.00 (1H, *m*, 6-CH), 4.09 (2H, *qd*, *J* = 7.1, 4.7 Hz, CH_2_CH_3_), 2.16 (3H, *s*, 8-CH_3_), 1.15 (3H, *t*, *J* = 7.1 Hz, CH_2_CH_3_). ^13^C NMR (101 MHz, DMSO-*d_6_*, *δ*): 168.8 (6-COOCH_2_CH_3_), 146.8 (C-4), 127.1 (C-7), 123.0 (C-8), 61.8 (3-COOCH_2_CH_3_), 57.1 (C-6), 16.2 (5-CH_3_), 14.3 (3-COOCH_2_CH_3_). UPLC-MS (ESI+): found *m/z* 210.0 [M + H]^+^; calculated for C_8_H_12_N_5_O_2_ 210.1.

#### 3.2.8. Synthesis of Ethyl 2,8-Dimethylimidazo[1,2-*b*]pyridazine-6-carboxylate **22**

A solution of ethyl 6-amino-5-methylpyridazine-3-carboxylate (65.3 g, 0.361 mol, 1 eq) in toluene (350 mL) was treated with finely powdered NaHCO_3_ (90.9 g, 1.083 mol, 3 eq) and anhydrous Na_2_SO_4_ (51.3 g, 0.361 mol, 1 eq). The resulting suspension was heated up to 85–90 °C with vigorous stirring and treated with the dropwise addition of chloroacetone (87 mL, 1.083 mol, 3 eq) dissolved in toluene (130 mL), accompanied by CO_2_ evolution. After the addition was complete, the resulting slurry was stirred at the same temperature for 3 h until the gas release stopped. Upon the completion of the reaction (TLC, eluent—EtOAc, *R_f_* = 0.13 for the starting compound, *R_f_* = 0.5 for the target product, UV-detection), the mixture was filtered through a TLC-grade silica gel pad (particle size 5–17 microns, ~85 g) and evaporated to dryness in vacuo. The residual dark-orange oil was dissolved in a minimal volume of the mixture of MTBE and *n*-C_6_H_14_ (5:1, by volume) and chilled to +4 °C for 12 h. The precipitate was filtered off, washed sequentially with 100 mL of an ice-cold mixture of MTBE and *n*-C_6_H_14_ (5:1, by volume) and *n*-C_6_H_14_ (2 × 30 mL) and dried at reduced pressure to a constant weight to give ethyl 2,8-dimethylimidazo[1,2-*b*]pyridazine-6-carboxylate as a light brown powder (67.2 g, 85%).

^1^H NMR (400 MHz, DMSO-*d_6_*, *δ*): 8.20 (1H, *d*, *J* = 1.0 Hz, 3-CH), 7.55 (1H, *q*, *J* = 1.1 Hz, 7-CH), 4.40 (2H, *q*, *J* = 7.1 Hz, CH_2_CH_3_), 2.60 (3H, *d*, *J* = 1.1 Hz, 8-CH_3_), 2.43 (3H, *d*, *J* = 0.8 Hz, 2-CH_3_), 1.36 (3H, *t*, *J* = 7.1 Hz, CH_2_CH_3_). ^13^C NMR (101 MHz, DMSO-*d_6_*, *δ*): 163.4 (6-COOCH_2_CH_3_), 145.3 (C-2), 142.4 (C-6), 139.6 (C-9), 136.3 (C-8), 116.0 (C-7), 115.4 (C-3), 62.3 (6-COOCH_2_CH_3_), 16.5 (8-CH_3_), 15.1 (2-CH_3_), 14.5 (3-COOCH_2_CH_3_). HRMS (ESI+): found *m/z* 220.1113 [M + H]^+^; calculated for C_11_H_14_N_3_O_2_ 220.1008, mp = 109–110 °C.

#### 3.2.9. Synthesis of 2,8-Dimethylimidazo[1,2-*b*]pyridazine-6-carboxylic Acid **13**

A vigorously stirred solution of NaOH (54 g, 1.35 mol, 5 eq) in 50% aqueous EtOH (500 mL) was treated with a slow addition of ethyl 2,8-dimethylimidazo[1,2-*b*]pyridazine-6-carboxylate (60.2 g, 0.270 mol, 1 eq) dissolved in 95% EtOH (350 mL), leading to the formation of a colloid. The latter was treated with deionized water (150 mL) and heated up to reflux with vigorous stirring for 2 h, under the UPLC-MS-monitoring. After the reaction was completed, EtOH was evaporated in vacuo and the residual aqueous solution was brought to pH = 3 with conc. H_2_SO_4_, accompanied with the precipitation of the target compound. The latter was filtered off, washed with 40 mL of cold deionized water, and air-dried. The traces of water were removed from the resulting 2,8-dimethylimidazo[1,2-*b*]pyridazine-6-carboxylic acid by co-evaporation with toluene, yielding the target product as a beige powder (47.2 g, 90%)

^1^H NMR (400 MHz, DMSO-*d_6_*, *δ*): 13.62 (1H, *br.s.*, 6-COOH), 8.14 (1H, *d*, *J* = 1.0 Hz, 3-CH), 7.53 (1H, *d*, *J* = 1.3 Hz, 7-CH), 2.58 (3H, *d*, *J* = 1.1 Hz, 8-CH_3_), 2.42 (3H, *d*, *J* = 0.8 Hz, 2-CH_3_). ^13^C NMR (101 MHz, DMSO-*d_6_*, *δ*): 164.9 (6-COOH), 145.0 (C-2), 143.3 (C-6), 139.6 (C-9), 136.0 (C-8), 116.2 (C-7), 115.3 (C-3), 16.5 (8-CH_3_), 15.1 (2-CH_3_). HRMS (ESI–): found *m/z* 190.0609 [M − H]^+^; calculated for C_9_H_8_N_3_O_2_ 190.0695, mp = 271–272 °C (dec).

#### 3.2.10. Synthesis of 2,8-Dimethylimidazo[1,2-*b*]pyridazine-6-carbonyl Chloride **23**

A vigorously stirred suspension of 2,8-dimethylimidazo[1,2-*b*]pyridazine-6-carboxylic acid (37.2 g, 0.195 mol, 1 eq) in anhydrous DCM (200 mL), was treated with dry DMF (2.2 mL, 30 mmol, 0.15 eq). A solution of oxalyl chloride (20.9 mL, 0.244 mol, 1.25 eq) in anhydrous DCM (30 mL) was added dropwise to the stirred reaction mixture at 20–22 °C. After the addition was complete, the reaction mixture was refluxed with stirring for 5 h until the completion of the reaction, cooled to 20–22 °C, and treated with DIPEA (33.9 mL, 0.195 mol, 1 eq) to convert the hydrochloride salt of the target compound into the free base. The resulting clear solution was used in the next step without isolation and purification.

#### 3.2.11. Synthesis of 5-((2,8-Dimethylimidazo[1,2-*b*]pyridazin-6-yl)(hydroxy)methylene)-2,2-dimethyl-1,3-dioxane-4,6-dione **24**

A stirred solution of freshly recrystallized Meldrum acid (31.7 g, 0.22 mol, 1.15 eq) and DMAP (4.88 g, 40 mmol, 0.2 eq) in anhydrous DCM (150 mL) was cooled to 0 °C. The dropwise addition of the solution from the previous step, containing 2,8-dimethylimidazo[1,2-b]pyridazine-6-carbonylchloride, led to the formation of a dark-red reaction mixture. The latter was stirred overnight at 20–22 °C, and the reaction was monitored by TLC (eluent—DCM:MeOH = 4:1 (*R_f_* = 0.5 for target product, ninhydrin visualization). After the completion of the reaction, the obtained solution was used in the next step without isolation and purification of the target product.

#### 3.2.12. Synthesis of 5-(Chloro(2,8-dimethylimidazo[1,2-*b*]pyridazin-6-yl)methylene)-2,2-dimethyl-1,3-dioxane-4,6-dione **25**

To a solution of 5-((2,8-dimethylimidazo[1,2-*b*]pyridazine-6-yl)(hydroxy)methylene)-2,2-dimethyl-1,3-dioxane-4,6-dione obtained on the previous step, an additional amount of DMAP (84.9 g, 0.696 mol; 4 eq based on triphosgene) was added under vigorous stirring. The resulting solution was cooled to 0 °C, and then a solution of triphosgene (51.5 g, 0.174 mol, 1 eq) in DCM (200 mL) was slowly added. The formation of an orange and then green suspension was observed. The resulting suspension was refluxed for 5 h, and the reaction was monitored by UPLC-MS (ESI+, *m/z* = 336.07, 338.06). After the reaction was completed, the resulting suspension was filtered through a pad of Celite^®^, and the precipitate was washed with DCM (2 × 50 mL). The filtrate containing the target 5-(chloro(2,8-dimethylimidazo[1,2-*b*]pyridazine-6-yl)methylene)-2,2-dimethyl-1,3-dioxane-4,6-dione was used in the next step without further purification.

#### 3.2.13. Synthesis of *tert*-Butyl 7-(6-(((2,2-dimethyl-4,6-dioxo-1,3-dioxan-5-ylidene)(2,8-dimethylimidazo[1,2-*b*]pyridazin-6-yl)methyl)amino)pyridin-3-yl)-4,7-diazaspiro[2.5]octane-4-carboxylate **26**

To a solution of 5-(chloro(2,8-dimethylimidazo[1,2-*b*]pyridazine-6-yl)methylene)-2,2-dimethyl-1,3-dioxane-4,6-dione obtained on the previous step, DIPEA (66.5 mL, 0.383 mol, 2.2 eq based on amine **28**) and a solution of *tert*-butyl 7-(6-aminopyridine-3-yl)-4,7-diazospiro[2.5]octane-4-carboxylate **28** (52.9 g, 0.174 mol, 0.9 eq based on acid **13**) in DCM (200 mL) were successively added. The reaction mixture was stirred at r.t. for 5 h, and the reaction was monitored by UPLC-MS (ESI+, *m/z* = 604.28, 605.21). After achieving full conversion of 5-(chloro(2,8-dimethylimidazo[1,2-*b*]pyridazine-6-yl)methylene)-2,2-dimethyl-1,3-dioxane-4,6-dione reagent by UPLC-MS, the reaction mixture was cooled, and the solvent was stripped down under reduced pressure. The residue was quenched with 800 mL of 25% aqueous NH_4_Cl_aq_ and 800 mL of EtOAc and filtered through the pad of Celite^®^. A filter cake was washed with a mixture of 100 mL of 25% aqueous NH_4_Cl_aq_ and 100 mL EtOAc, yielding a clear bi-phasic filtrate. The organic phase was separated; the aqueous layer was additionally washed with EtOAc (2 × 200 mL). The combined organic layers were washed with brine (250 mL), dried over anhydrous Na_2_SO_4_ for 1 h under vigorous stirring, and filtered off. The solution was evaporated to dryness to give technically pure *tert*-butyl 7-(6-(((2,2-dimethyl-4,6-dioxo-1,3-dioxan-5-ylidene)(2,8-dimethylimidazo[1,2-*b*]pyridazin-6-yl)methyl)amino)pyridin-3-yl)-4,7-diazaspiro[2.5]octane-4-carboxylate as dark-green crystal powder (83.9 g, 80% on the added amine **28**).

^1^H NMR (400 MHz, DMSO-*d_6_*, *δ*): 12.49 (1H, *s*, NH), 7.93 (1H, *s*, 3″-CH), 7.75 (1H, *d*, *J* = 2.9 Hz, 2′-CH), 7.22 (1H, *dd*, *J* = 9.1, 3.0 Hz, 5′-CH), 7.14 (1H, *d*, *J* = 1.4 Hz, 7″-CH), 6.87 (1H, *d*, *J* = 9.0 Hz, 4′-CH), 3.50–3.47 (2H, *m*, 6-CH_2_), 3.08-3.06 (2H, *m*, 5-CH_2_), 2.95 (2H, *s*, 8-CH_2_), 2.50 (3H, *s*, 8″-CH_3_), 2.37 (3H, *s*, 2″-CH_3_), 1.77 (3H, *s*, (CH_3_)C(O)(O)(CH_3_)), 1.73 (3H, *s*, (CH_3_)C(O)(O)(CH_3_)), 1.38 (9H, *s*, -C(O)C(CH_3_)_3_), 0.88-0.85 (2H, *m*, 1-CH_2_, 2-CH_2_), 0.77-0.74 (2H, *m*, 1-CH_2_, 2-CH_2_). ^13^C NMR (101 MHz, DMSO-*d_6_*, *δ*): 165.5 (C=C(=NH)-), 161.2 (2C, -C(=O)-), 160.7 (C-6″), 154.6 (-C(O)C(CH_3_)_3_), 146.7 (C-6′), 142.8 (C-2″), 140.4 (C-2′), 138.4 (C-9″), 134.7 (C-2′), 134.6 (C-8′), 123.5 (C-5′), 117.1 (C-4′), 116.9 (C-7″), 114.6 (C-3″), 103.5 (C=C(=NH)-), 87.9 ((CH_3_)C(O)(O)(CH_3_)), 79.1 (-C(O)C(CH_3_)_3_), 53.3 (C-8), 45.9 (C-5), 44.9 (C-6), 37.2 (C-3), 27.9 (-C(O)C(CH_3_)_3_), 26.2 (-C(=O)-), 26.2 (-C(=O)-), 16.0 (8″-CH_3_), 14.4 (2″-CH_3_), 13.5 (C-1, C-2). UPLC-MS (ESI+): found *m/z* 604.2 [M + H]^+^; calculated for C_31_H_38_N_7_O_6_ 604.3.

#### 3.2.14. Synthesis of 7-(4-(*tert*-Butoxycarbonyl)-4,7-diazaspiro[2.5]octan-7-yl)-2-(2,8-dimethylimidazo[1,2-*b*]pyridazin-6-yl)-4-oxo-4*H*-pyrido[1,2-*a*]pyrimidine-3-carboxylic Acid **27**

The crude *tert*-butyl 7-(6-(((2,2-dimethyl-4,6-dioxo-1,3-dioxane-5-ylidene)(2,8-dimethylimidazo[1,2-*b*]pyridazine-6-yl)methyl)amino)pyridine-3-yl)-4,7-diazospiro[2.5]octane-4-carboxylate **26** (80 g, 0.133 mol) obtained at the previous stage was dissolved in *n*-BuOH (480 mL) under vigorous stirring, and then the reaction mixture was heated at 95 °C for 6 h. The reaction was monitored by UPLC-MS (ESI+, *m/z* = 546.24). After achieving full conversion of *tert*-butyl 7-(6-(((2,2-dimethyl-4,6-dioxo-1,3-dioxane-5-ylidene)(2,8-dimethylimidazo[1,2-*b*]pyridazine-6-yl)methyl)amino)pyridine-3-yl)-4,7-diazospiro[2.5]octane-4-carboxylate, the reaction mixture was cooled to 60 °C.

To obtain an analytically pure *7*-(4-(tert-butoxycarbonyl)-4,7-diazaspiro[2.5]octan-7-yl)-2-(2,8-dimethylimidazo[1,2-*b*]pyridazin-6-yl)-4-oxo-4*H*-pyrido[1,2-*a*]pyrimidine-3-carboxylic acid sample, an aliquot of the resulting reaction mixture was evaporated to dryness. The residue was purified by column chromatography on silica gel to obtain the target compound (50 mg) (eluent—DCM:MeOH (gravity elution in a gradient from 100:0 to 85:15, by volume). The remaining reaction mixture was used in the final stage of synthesis without additional treatment and purification.

^1^H NMR (400 MHz, CDCl_3_, *δ*): 8.41 (1H, *d*, *J* = 2.5 Hz, 6-CH), 7.85-7.79 (2H, *m*, 8-CH, 9-CH), 7.64 (1H, *d*, *J* = 0.9 Hz, 3′-CH), 6.92 (1H, *d*, *J* = 1.2 Hz, 7′-CH), 3.73 (2H, *dd*, *J* = 6.1, 4.3 Hz, 6″-CH_2_), 3.32 (2H, *t*, *J* = 5.2 Hz, 5″-CH_2_), 3.10 (2H, *s*, 8″-CH_2_), 2.61 (3H, *d*, *J* = 1.1 Hz, 8′-CH_3_), 2.45 (3H, *d*, *J* = 0.9 Hz, 2′-CH_3_), 1.43 (9H, *s*, -C(O)C(CH_3_)_3_), 1.10–1.07 (2H, *m*, 1″-CH_2_, 2″-CH_2_), 0.86–0.83 (2H, *m*, 1″-CH_2_, 2″-CH_2_). ^13^C NMR (101 MHz, CDCl_3_, *δ*): 164.0 (3-COOH), 161.5 (C-4), 160.4 (C-10), 155.2 (-C(O)C(CH_3_)_3_), 151.2 (C-6′), 145.3 (C-2), 144.0 (C-7), 143.2 (C-2′), 139.3 (C-9′), 135.4 (C-8′), 132.5 (C-8), 127.3 (C-9), 116.5 (C-7′), 114.6 (C-3′), 109.7 (C-6), 102.5 (C-3), 80.8 (-C(O)C(CH_3_)_3_), 54.4 (C-8″), 46.9 (C-5″), 44.9 (C-6″), 37.1 (C-3″), 28.4 (-C(O)C(CH_3_)_3_), 16.8 (8′-CH_3_), 14.7 (2′-CH_3_), 14.3 (C-1″, C-2″). HRMS (ESI+): found *m/z* 546.2475 [M + H]^+^; calculated for C_28_H_32_N_7_O_5_ 546.2387, mp = 185–186 °C (dec).

#### 3.2.15. Synthesis of 2-(2,8-Dimethylimidazo[1,2-*b*]pyridazin-6-yl)-7-(4,7-diazaspiro[2.5]octan-7-yl)-4*H*-pyrido[1,2-*a*]pyrimidin-4-one (Risdiplam)

To the solution of AcCl (96.8 mL, 1.365 mol, 7 eq (on starting acid **13**) in *n*-butanol (250 mL) the previously obtained solution of 7-(4-(*tert*-butoxycarbonyl)-4,7-diazaspiro[2.5]octan-7-yl)-2-(2,8-dimethylimidazo[1,2-*b*]pyridazin-6-yl)-4-oxo-4*H*-pyrido[1,2-*a*]pyrimidine-3-carboxylic acid was added dropwise at 60 °C. The resulting mixture was heated at 105°C for 12–16 h. After several hours, the formation of a yellow precipitate was observed, and the reaction was monitored by UPLC-MS (ESI+, *m/z* = 402.20). After the reaction was completed, the precipitate was filtered, washed with cooled *n*-butanol (3 × 100 mL), and successively dried in air and in high vacuum to obtain the corresponding dihydrochloride 2-(2,8-dimethylimidazo[1,2-*b*]pyridazine-yl)-7-(4,7-diazospiro[2.5]octane-7-yl)-4*H*-pyrido[1,2-*a*]pyrimidine-4-one in the form of a yellow powder (51.4 g, 55.4% (from starting 2,8-dimethylimidazo[1,2-*b*]pyridazine-6-carboxylic acid)).

Next, the resulting dihydrochloride (51.4 g, 0.108 mol, 1 eq) was dissolved in deionized water (150 mL) under vigorous stirring, and ethanol (150 mL) was added. An aqueous NaOH solution (13.06 mL, 1.349 g/mL, 32%, 0.141 mol, 1.3 eq) was added drop by drop to the resulting solution until pH = 13, and yellow precipitation was observed. The resulting suspension was heated at 50 °C for 6 h under vigorous stirring, and then cooled to r.t. for 1 h, the precipitate was filtered and washed with H_2_O:EtOH = 2:1 mixture. The resulting precipitate was dried at reduced pressure at 50 °C to a constant mass with the formation of trihydrate 2-(2,8-dimethylimidazo[1,2-*b*]pyridazine-6-yl)-7-(4,7-diazospiro[2.5]octane-7-yl)-4*H*-pyrido[1,2-*a*]pyrimidine-4-one (43.5 g, 80%; Overall yield starting from acid **13**—44.3%).

^1^H NMR (400 MHz, CDCl_3_, *δ*): 8.36 (1H, *d*, *J* = 2.4 Hz, 6-CH), 7.84 (1H, *q*, *J* = 1.1 Hz, 7′-CH), 7.72 (1H, *d*, *J* = 1.0 Hz, 3′-CH), 7.66–7.63 (1H, *m*, 9-CH), 7.63–7.59 (1H, *m*, 8-CH), 7.29 (1H, *s*, 3-CH), 3.19–3.16 (2H, *m*, 6″-CH_2_), 3.12–3.09 (2H, *m*, 5″-CH_2_), 3.00 (2H, *s*, 8″-CH_2_), 2.65 (3H, *d*, *J* = 1.1 Hz, 8′-CH_3_), 2.47 (3H, *d*, *J* = 0.8 Hz, 2′-CH_3_), 0.69–0.66 (2H, *m*, 1″-CH_2_, 2″-CH_2_), 0.59–0.56 (2H, *m*, 1″-CH_2_, 2″-CH_2_). ^13^C NMR (101 MHz, CDCl_3_, *δ*): 158.1 (C-4), 156.2 (C-6′), 148.5 (C-2), 147.2 (C-10), 144.1 (C-2′), 142.2 (C-7), 140.0 (C-9′), 135.6 (C-8′), 131.2 (C-8), 126.7 (C-9), 114.9 (C-7′), 114.7 (C-3′), 110.0 (C-6), 99.2 (C-3), 56.6 (C-8″), 49.9 (C-5″), 44.5 (C-6″), 36.5 (C-3″), 16.9 (8′-CH_3_), 14.9 (2′-CH_3_), 13.0 (C-1″, C-2″). HRMS (ESI+): found *m/z* 402.2051 [M + H]^+^; calculated for C_22_H_24_N_7_O 402.1964, mp = 269–270 °C (dec).

#### 3.2.16. Synthesis of *tert*-Butyl 4,7-diazaspiro[2.5]octane-4-carboxylate (Oxalate Salt) **30**

TrtCl (153 g, 0.549 mol, 1.15 eq) and Et_3_N (100 mL, 0.716 mol, 1.5 eq) were added portionwise to a stirred solution of 4,7-diazospiro[2.5]octane **29** (53.4 g, 0.477 mol, 1 eq) in MeCN (200 mL). The reaction mixture was stirred at 20–22 °C for 8 h. After completion of the reaction (TLC, eluent—DCM:MeOH = 9:1, by volume, *R_f_* = 0.45 for target product, UV-detection), the precipitate was filtered off, washed with 100 mL of MeCN, and suspended in 200 mL of THF. The resulting slurry was filtered through a pad of Celite^®^, and stripped down from the solvent in vacuo, to yield the target 7-trityl-4,7-diazospiro[2,5]octane as a viscous yellow oil (143.7 g, 85%), which slowly crystallizes upon standing at 20–22 °C. Obtained 7-trityl-4,7-diazospiro[2,5]octane was redissolved in THF (350 mL) and quenched with Et_3_N (96 mL, 0.690 mol, 1.7 eq). A solution of Boc_2_O (115.1 g, 0.528 mol, 1.3 eq) in THF (350 mL) was added to the resulting solution with vigorous stirring. The reaction mixture was stirred at 20–22 °C for 8 h and after the completion of the reaction (TLC, eluent—DCM:MeOH = 9:1, by volume, with a drop of Et_3_N, *R_f_* = 0.45 for 7-trityl-4,7-diazaspiro[2,5]octane, UV-detection), the mixture was quenched with 5% aqueous NaHCO_3_ solution (500 mL). The organic layer was separated and the aqueous layer was extracted with MTBE (2 × 250 mL). The combined organic layers were washed with 250 mL of brine, dried over anhydrous Na_2_SO_4_ with vigorous stirring for 1 h, filtered, and evaporated to dryness in vacuo to yield *tert*-butyl 7-trityl-4,7-diazospiro[2,5]octane-4-carboxylate as a viscous yellow oil (175.2 g, 95%).

The latter was dissolved in MTBE (400 mL) and a solution of oxalic acid (104.2 g, 1.158 mol, 3 eq) in a mixture of THF (200 mL), and MTBE (200 mL) was added. The resulting mixture was heated up to 45 °C with constant stirring for 12 h and then cooled to 20–22 °C. The precipitate formed during the reaction was filtered off. The filter cake was washed with MTBE (2 × 150 mL) and dried at reduced pressure to a constant weight yielding the target *tert*-butyl 4,7-diazaspiro[2.5]octane-4-carboxylate hydrogen oxalate **30** as a white powder (70 g, 60%)

^1^H NMR (400 MHz, CDCl_3_, *δ*): 3.52–3.50 (2H, *m*, 5-CH_2_), 2.87–2.85 (2H, *m*, 6-CH_2_), 2.67 (2H, *s*, 8-CH_2_), 1.48 (9H, *s*, -C(O)C(CH_3_)_3_), 0.99–0.96 (2H, *m*, 1-CH_2_, 2-CH_2_), 0.75–0.71 (2H, *m*, 1-CH_2_, 2-CH_2_). ^13^C NMR (101 MHz, CDCl_3_, *δ*): 155.7 (-C(O)C(CH_3_)_3_), 79.7 (-C(O)C(CH_3_)_3_), 52.4 (C-8), 48.3 (C-5), 45.6 (C-6), 39.8 (C-3), 28.4 (-C(O)C(CH_3_)_3_), 13.6 (C-1, C-2). HRMS (ESI+): found *m/z* 213.1622 [M + H]^+^; calculated for C_11_H_21_N_2_O_2_ 213.1525, mp = 55–56 °C.

#### 3.2.17. Synthesis of *tert*-Butyl 7-(6-nitropyridin-3-yl)-4,7-diazaspiro[2.5]octane-4-carboxylate **31**

A solution of *tert*-butyl 4,7-diazaspiro[2.5]octane-4-carboxylate oxalate salt **30** (86.4 g, 0.286 mol, 1.05 eq), 5-bromo-2-nitropyridine (55.2 g, 0.272 mol, 1 eq), DBU (144.7 g, 0.952 mol, 3.5 eq) and LiCl (46.2 g, 1.088 mol, 4 eq) in dry DMSO (400 mL) was stirred at 85 °C for 36 h. After the completion of the reaction (TLC, eluent—*n*-C_6_H_14_:EtOAc = 2:1, by volume, *R_f_* = 0.45 for 5-bromo-2-nitropyridine), the resulting solution was cooled to 20–22 °C, poured into deionized water (1 L) and the obtained precipitate was filtered. The filter cake was washed with deionized water (2 × 100 mL), and dissolved in DCM (2.5 L). The resulting solution was dried over anhydrous Na_2_SO_4_ for 1 h with vigorous stirring, filtered, and stripped down from the solvent in vacuo. The residue was redissolved in 4 L of EtOAc and precipitated with 16 L of *n*-C_6_H_14_. The resulting yellow powder was filtered off, washed with *n*-C_6_H_14_ (500 mL), and dried at reduced pressure to a constant weight to yield *tert*-butyl 7-(6-nitropyridin-3-yl)-4,7-diazaspiro[2.5]octane-4-carboxylate **31** (63.6 g, 70%).

^1^H NMR (400 MHz, DMSO-*d_6_*, *δ*): 8.23 (1H, *d*, *J* = 3.0 Hz, 2′-CH), 8.15 (1H, *d*, *J* = 9.2 Hz, 5′-CH), 7.46 (1H, *dd*, *J* = 9.3, 3.1 Hz, 4′-CH), 3.59 (2H, *dd*, *J* = 6.5, 3.9 Hz, 5-CH_2_), 3.50 (2H, *dd*, *J* = 6.6, 3.9 Hz, 6-CH_2_), 3.36 (2H, *s*, 8-CH_2_), 1.42 (9H, *s*, -C(O)C(CH_3_)_3_), 0.98–0.95 (2H, *m*, 1-CH_2_, 2-CH_2_), 0.88–0.85 (2H, *m*, 1-CH_2_, 2-CH_2_). ^13^C NMR (101 MHz, DMSO-*d_6_*, *δ*): 155.1 (-C(O)C(CH_3_)_3_), 150.6 (C-3′), 147.3 (C-6′), 133.9 (C-2′), 121.0 (C-4′), 120.2 (C-5′), 79.9 (-C(O)C(CH_3_)_3_), 52.7 (C-8), 48.5 (C-5), 45.3 (C-6), 37.8 (C-3), 28.5 (-C(O)C(CH_3_)_3_), 14.0 (C-1, C-2). HRMS (ESI+): found *m/z* 335.1739 [M + H]^+^; calculated for C_16_H_23_N_4_O_4_ 335.1641, mp = 162–163 °C.

#### 3.2.18. Synthesis of *tert*-Butyl 7-(6-aminopyridin-3-yl)-4,7-diazaspiro[2.5]octane-4-carboxylate **28**

*tert*-Butyl 7-(6-nitropyridin-3-yl)-4,7-diazaspiro[2.5]octane-4-carboxylate **31** (63.6 g, 0.190 mol, 1 eq) was hydrogenated for 8 h at ordinary pressure in EtOAc (3.1 L) in the presence of 15 g of 10% Pd on carbon with constant stirring. After the completion of the starting material (TLC, eluent—DCM:MeOH = 9:1, by volume, *R_f_* = 0.45 for the starting material), the resulting suspension was filtered through a pad of Celite^®^ and the filtrate was evaporated to dryness in vacuo, leaving the technical grade *tert*-butyl 7-(6-aminopyridin-3-yl)-4,7-diazaspiro[2.5]octane-4-carboxylate (54.9 g, 95%) in the residue as light-orange oil.

^1^H NMR (400 MHz, DMSO-*d_6_*, *δ*): 7.57 (1H, *dd*, *J* = 3.0, 0.7 Hz, 2′-CH), 7.14 (1H, *dd*, *J* = 8.9, 3.0 Hz, 4′-CH), 6.39 (1H, *dd*, *J* = 8.9, 0.7 Hz, 5′-CH), 5.40 (2H, *br.s.*, 6′-NH_2_), 3.54 (2H, *dd*, *J* = 6.1, 3.9 Hz, 5-CH_2_), 2.89 (2H, *dd*, *J* = 6.3, 3.6 Hz, 6-CH_2_), 2.74 (2H, *s*, 8-CH_2_), 1.41 (9H, *s*, -C(O)C(CH_3_)_3_), 0.93–0.90 (2H, *m*, 1-CH_2_, 2-CH_2_), 0.83–0.80 (2H, *m*, 1-CH_2_, 2-CH_2_). ^13^C NMR (101 MHz, DMSO-*d_6_*, *δ*): 154.9 (C-6′), 155.2 (-C(O)C(CH_3_)_3_), 139.2 (C-3′), 136.5 (C-2′), 128.6 (C-4′), 108.7 (C-5′), 79.4 (-C(O)C(CH_3_)_3_), 57.2 (C-8), 49.8 (C-6), 46.0 (C-5), 38.2 (C-3), 28.5 (-C(O)C(CH_3_)_3_), 14.1 (C-1, C-2). HRMS (ESI+): found *m/z* 305.1974 [M + H]^+^; calculated for C_16_H_25_N_4_O_2_ 305.1899.

## 4. Conclusions

The brand-new effective protocol for the multigram scale synthesis of 2,8-dimethylimidazo[1,2-*b*]pyridazine-6-carboxylic acid **13** has been developed. The clear advantage of the proposed protocol over alternative methods is the highly efficient preparation of all intermediates, characterized by their crystallinity, ease of isolation, and stability. Equally important is the ability to perform all described synthetic transformations without relying on metal-catalyzed cross-coupling reactions, using simple and readily available starting materials, while achieving high yields of the target compound **13** (46% over 6 steps (from 5-methyl-6-oxo-1,6-dihydropyridazine-3-carboxylic acid **17**)) without the need for chromatographic purification. Also, the developed method offers a cost-effective alternative to the commercial acquisition of compound **17**, allowing its synthesis on a hundred-gram scale from low-cost methyl pyruvate (~$30/kg), considering its high market price (>$1000/g, Enamine). The proposed production method enables the synthesis of not only valuable building blocks **13** and **17** for the needs of medicinal chemistry but also provides a direct pathway to the target molecule risdiplam. Moreover, we have improved the known scheme for the risdiplam synthesis from acid **13** by creating a checkpoint inside the one-pot method, which makes it possible to obtain a target substance with high purity, robustness, and a good yield.

## Figures and Tables

**Figure 1 molecules-30-03011-f001:**
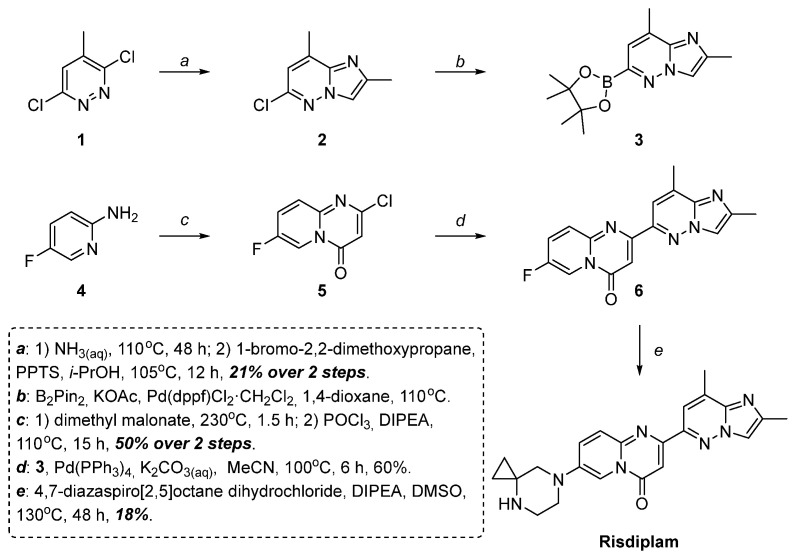
The pilot scheme of risdiplam synthesis [13].

**Figure 2 molecules-30-03011-f002:**
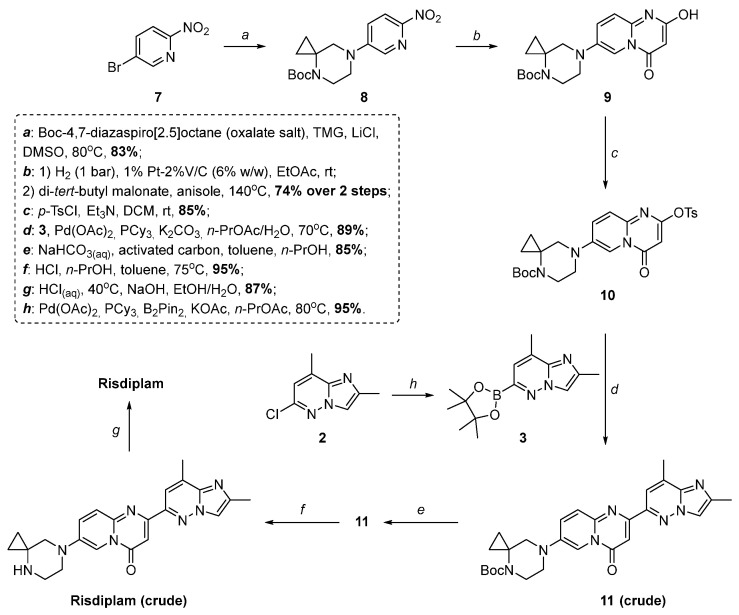
Commercial manufacturing route for risdiplam [14].

**Figure 3 molecules-30-03011-f003:**
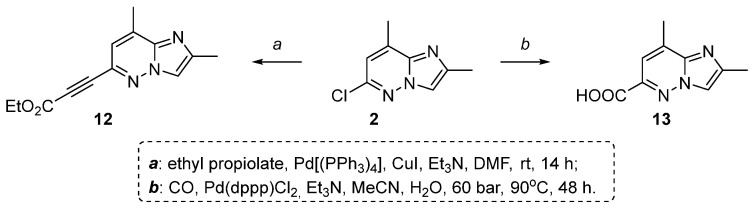
Alternative 2,8-dimethylimidazo[1,2-*b*]pyridazine derivatives as building blocks in the risdiplam synthesis [16,17].

**Figure 4 molecules-30-03011-f004:**
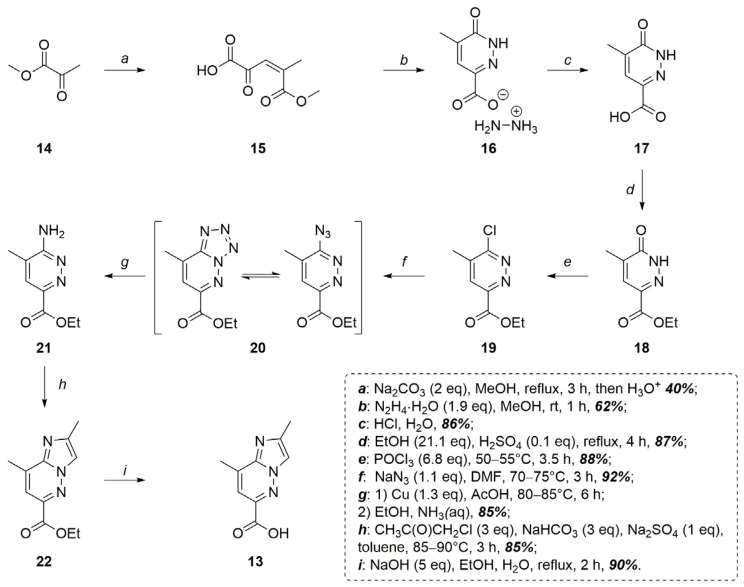
The developed 2,8-dimethylimidazo[1,2-*b*]pyridazine-6-carboxylic acid **13** effective synthetic route.

**Figure 5 molecules-30-03011-f005:**
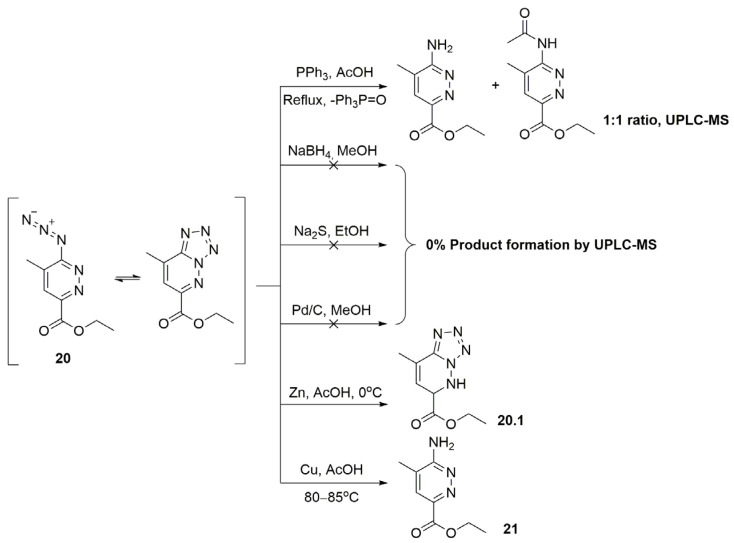
Selection of the compound **20** reduction conditions.

**Figure 6 molecules-30-03011-f006:**
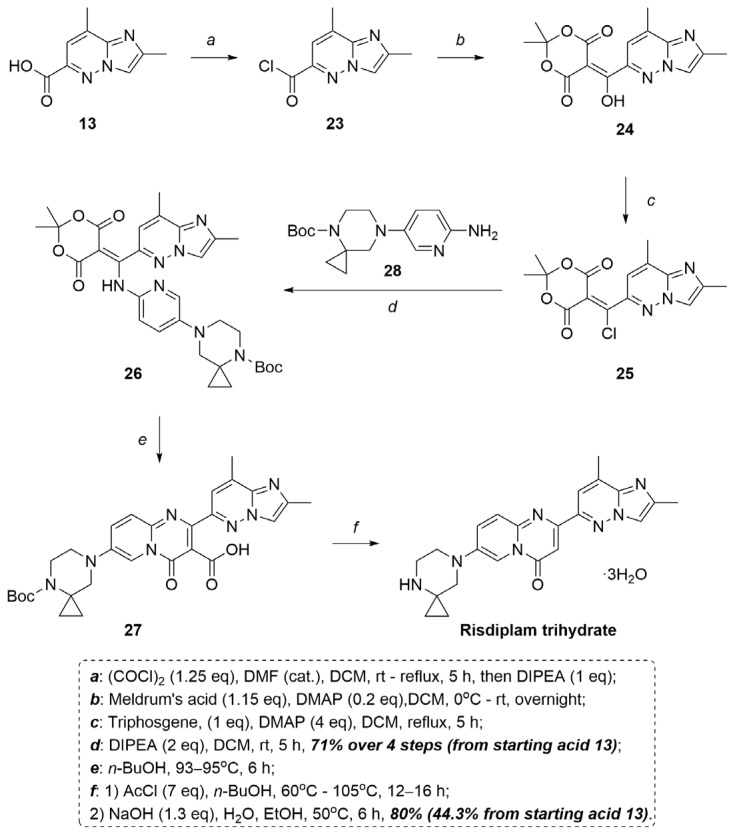
Risdiplam trihydrate synthesis.

**Figure 7 molecules-30-03011-f007:**
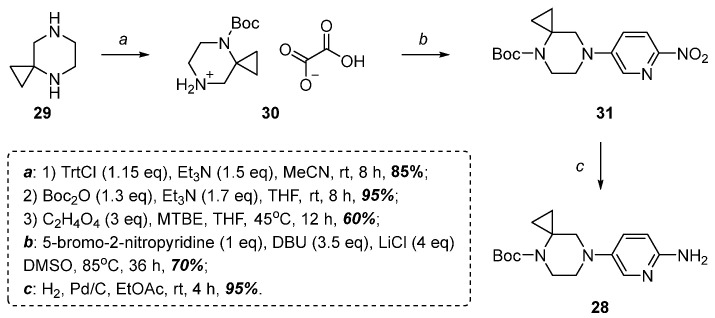
*tert*-Butyl 7-(6-aminopyridin-3-yl)-4,7-diazaspiro[2.5]octane-4-carboxylate **28** synthetic route.

## Data Availability

Data are contained within the article and Appendix A.

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
