# Peer review of "A Brand-New Metal Complex Catalyst-Free Approach to the Synthesis of 2,8-Dimethylimidazo[1,2-*b*]pyridazine-6-Carboxylic Acid—A Key Intermediate in Risdiplam Manufacturing Process"

_molecules, 2025, doi:10.3390/molecules30143011_

Round 1

Reviewer 1 Report

Comments and Suggestions for Authors

In manuscript “A brand-new metal complex catalyst-free approach to the synthesis of 2,8-dimethylimidazo[1,2-b]pyridazine-6-carboxylic acid – a key intermediate in risdiplam manufacturing process,”  the authors describe a novel synthetic route of Risdiplam which involves the use of  2,8-dimethylimidazo[1,2-b]pyridazine-6-carboxylic acid (compound 13) as the key synthetic intermediate. I recommend the publication of this manuscript in Molecules after major revision.

A first suggestion is to make the introduction more concise. In this sense, it is not clear from reading the first few pages whether compound 13 has been used in a previous synthetic route or whether the objective is to propose compound 13 as a key compound for a new synthesis. In the latter case, it is suggested to place a retrosynthetic analysis at the end of the introduction or at the beginning of the results and discussion section. It is also convenient to discuss the spectroscopic characterization of compound 13 as well as compound 27 with emphasis on the spectroscopic signals that allow the identification of the 3-membered ring.

The synthetic procedure of compound 13 should be included in manuscript.

Minimal corrections: on page 3, lines 85 and 90, mark in black the numbers for compounds 6 and 10.

Reviewer 2 Report

Comments and Suggestions for Authors

The manuscript entitled «A brand-new metal complex catalyst-free approach to the synthesis of 2,8-dimethylimidazo[1,2-b]pyridazine-6-carboxylic acid – a key intermediate in risdiplam manufacturing process» by Georgiy Korenev et al. present a new and an interesting way of a pyridazine-6-carboxylic acid derivative.

Remarks and suggestions:

  1. According to the results of iThenticate system, the presented manuscript has 35% of text plagiarism. The manuscript should be carefully checked.
  2. The title is a bit too long.
  3. In the Discussion, add more specific advantages of the proposed acid synthesis pathway (e.g., yield and total cost ($)) compared to the previously known method. Add corresponding results to Abstract and / or Conclusions.
  4. Compound #15 was synthesized previously in the exact same manner. Add corresponding citations (for instance, doi: 1021/acsomega.0c00877 ).
  5. Carefully check 1H NMR spectra (compounds #17, 18) for missing protons.
  6. Add high-quality HRMS spectra figures to the Supplementary.
  7. The melting point of compound #17 differs by more than 10 oC from that described in the literature (doi: 1021/ja01583a043 ).
  8. Add an IR spectrum of #20 to confirm that the molecule in solid form exists in an azido form.

Round 2

Reviewer 1 Report

Comments and Suggestions for Authors

After revision of revised manuscript “A brand-new metal complex catalyst-free approach to the synthesis of 2,8-dimethylimidazo[1,2-b]pyridazine-6-carboxylic acid – a key intermediate in risdiplam manufacturing process,” authors have made suggested changes and now manuscript is suitable for publication in its current form.

Author Response

We would like to once again express our sincere gratitude to the reviewers for their valuable comments, thoughtful questions, and the positive assessment of the revised version. Your constructive feedback has played a crucial role in enhancing the scientific quality and improving the overall clarity of the manuscript.

Reviewer 2 Report

Comments and Suggestions for Authors

All my comments and remarks on the manuscript were taken into account. Therefore, the manuscript can be considered for publication in its current form.

Author Response

(The authors gave the same response as above.)
